# PUFA Supplementation and Heart Failure: Effects on Fibrosis and Cardiac Remodeling

**DOI:** 10.3390/nu13092965

**Published:** 2021-08-26

**Authors:** Francesca Oppedisano, Rocco Mollace, Annamaria Tavernese, Micaela Gliozzi, Vincenzo Musolino, Roberta Macrì, Cristina Carresi, Jessica Maiuolo, Maria Serra, Antonio Cardamone, Maurizio Volterrani, Vincenzo Mollace

**Affiliations:** 1Department of Health Sciences, Institute of Research for Food Safety & Health (IRC-FSH), University Magna Graecia, 88100 Catanzaro, Italy; rocco.mollace@gmail.com (R.M.); an.tavernese@gmail.com (A.T.); micaela.gliozzi@gmail.com (M.G.); xabaras3@hotmail.com (V.M.); robertamacri85@gmail.com (R.M.); carresi@unicz.it (C.C.); jessicamaiuolo@virgilio.it (J.M.); mariaserra@gmail.com (M.S.); tony.c@outlook.it (A.C.); 2Nutramed S.c.a.r.l., Complesso Ninì Barbieri, Roccelletta di Borgia, 88021 Catanzaro, Italy; 3Division of Cardiology, University Hospital Policlinico Tor Vergata, 00133 Rome, Italy; 4Department of Cardiology, IRCCS San Raffaele Pisana, 00166 Rome, Italy; maurizio.volterrani@sanraffaele.it

**Keywords:** PUFAs, heart failure, HFrEF, HFpEF, myocardial infarction (MI), cardiac remodeling, cardiac fibrosis, inflammation, cardioprotective mechanism

## Abstract

Heart failure (HF) characterized by cardiac remodeling is a condition in which inflammation and fibrosis play a key role. Dietary supplementation with n-3 polyunsaturated fatty acids (PUFAs) seems to produce good results. In fact, eicosapentaenoic acid (EPA) and docosahexaenoic acid (DHA) have anti-inflammatory and antioxidant properties and different cardioprotective mechanisms. In particular, following their interaction with the nuclear factor erythropoietin 2 related factor 2 (NRF2), the free fatty acid receptor 4 (Ffar4) receptor, or the G-protein coupled receptor 120 (GPR120) fibroblast receptors, they inhibit cardiac fibrosis and protect the heart from HF onset. Furthermore, n-3 PUFAs increase the left ventricular ejection fraction (LVEF), reduce global longitudinal deformation, E/e ratio (early ventricular filling and early mitral annulus velocity), soluble interleukin-1 receptor-like 1 (sST2) and high-sensitive C Reactive protein (hsCRP) levels, and increase flow-mediated dilation. Moreover, lower levels of brain natriuretic peptide (BNP) and serum norepinephrine (sNE) are reported and have a positive effect on cardiac hemodynamics. In addition, they reduce cardiac remodeling and inflammation by protecting patients from HF onset after myocardial infarction (MI). The positive effects of PUFA supplementation are associated with treatment duration and a daily dosage of 1–2 g. Therefore, both the European Society of Cardiology (ESC) and the American College of Cardiology/American Heart Association (ACC/AHA) define dietary supplementation with n-3 PUFAs as an effective therapy for reducing the risk of hospitalization and death in HF patients. In this review, we seek to highlight the most recent studies related to the effect of PUFA supplementation in HF. For that purpose, a PubMed literature survey was conducted with a focus on various in vitro and in vivo studies and clinical trials from 2015 to 2021.

## 1. Introduction

Heart failure (HF) arises as a result of various cardiovascular diseases and leads to numerous cases of hospitalization [1,2,3]. It is known that in HF, there is an alteration of the ventricular filling or of the ejection of blood. To this condition are added symptoms such as dyspnea and fatigue as well as pulmonary and peripheral edema [4]. There are many comorbidities associated with heart failure, such as hypertension, diabetes, and obesity or hyperlipidemia, which are responsible for the increased hospitalization and mortality of patients with HF, even of relatively young subjects [5,6,7,8,9,10]. In addition, there has been an increase in HF cases linked to the ageing population [5,11]. According to Katsi V. et al. 2017, HF patients are categorized on the basis of underlying left ventricular ejection fraction (LVEF) and are mainly distinguished as patients with HF with a reduced ejection fraction (HFrEF) and patients with HF with a preserved ejection fraction (HFpEF) [12]; the latter represents more than 50% of HF cases and has a much less well-understood pathophysiology than HFrEF [2,12,13,14,15]. HF is characterized by cardiac remodeling that usually occurs following events related to hypertension, myocardial infarction (MI), myocarditis, or heart valve disease and is a condition in which inflammation and fibrosis are determinants [3,16] (Figure 1).

HF is characterized by cardiac remodeling caused by cardiac inflammation and fibrosis; it is distinguished in both HF with a reduced ejection fraction (HFrEF) and HF with a preserved ejection fraction (HFpEF). 

In particular, inflammation determines the presence of immune cells in the heart tissue and can be distinguished by sterile myocardial inflammation and non-sterile myocardial inflammation [3]. In order to resolve myocardial inflammation, anti-inflammatory processes are activated, which result in the onset of a pro-fibrotic state. In myocardial fibrosis, extracellular matrix proteins are more abundant and show structural and functional alterations. Both cardiac inflammation and fibrosis are being studied in numerous animal models, as they could be the target of new heart failure therapies [3]. Experimental and clinical studies conducted over the years in patients with acute and chronic heart failure have demonstrated the involvement of both the innate and adaptive immune systems in the disease [17,18,19]. In 2019, the CANTOS (Canakinumab Anti-Inflammatory Thrombosis Outcomes Study) trial, using anti-cytokine therapy with a monoclonal antibody against IL-1β, established that chronic inflammation plays an important role in the pathogenesis of heart failure. Therefore, anti-inflammatory therapies could be administered to heart failure patients with a cardio-inflammatory phenotype [18]. From a pathophysiological point of view, HF has a reduced ATP (adenosine triphosphate) availability, changes in Na^+^ and Ca^2+^ handling, and myocytes characterized by oxidative stress. In all of these processes, the mitochondria are involved and therefore play a fundamental role in HF. In fact, the altered mitochondrial structure, dynamics, and function in HF generate an energetic deficit and oxidative stress [2,20,21,22,23,24,25,26,27,28,29]. In particular, the process known as ROS-induced ROS release is triggered. In this condition, the ROS release from the mitochondria triggers a further release of ROS from neighboring mitochondria; in this way, high levels of ROS are present, which alter the excitation–contraction coupling, generating cardiac remodeling and cell death [2,20,30,31]. It has been shown that, in the early stages of HF, oxidative stress and mitochondrial dysfunction determine the onset of a condition known as “cardiomyocyte frailty”, in which myocardial cells undergo apoptosis if there are no protective mechanisms such as autophagy and the overexpression of endogenous antioxidant enzymes [5]. There is also altered function of the endoplasmic reticulum, reduced nitric oxide (NO) release, metalloproteinase (MMP) dysregulation, and a change in cardiac stem cell mobilization. All of this determines an increase in the left ventricle’s stiffness, generating diastolic dysfunction [5]. HF can also arise as a result of a pathological increase in the heart size [32]. Following MI, both the infarcted and non-infarcted areas present ventricular morphological and functional changes, such as greater ventricular preload and afterload and lower systolic volume and cardiac output, conditions that determine HF onset. In order to restore the physiological condition, a neuro-humoral response is activated, which is characterized by a greater sympathetic discharge and hormone synthesis such as angiotensin II and aldosterone, thus increasing peripheral vascular resistance. In the early stages of HF, this represents a compensatory mechanism, but in the long term, it leads to a worsening of symptoms [33]. Regarding the therapy for HFrEF, clinical trials have demonstrated the efficacy of several therapies evaluated at the same time. To date, therapies include treatment with vasodilators, angiotensin-converting enzyme (ACE) inhibitors, and angiotensin receptor blockers (ARBs) as well as beta-blockers and mineralocorticoid receptor antagonists (MRAs) [34,35,36]. In addition, the US Food and Drug Administration (FDA) has recently approved the sodium–glucose co-transporter 2 (SGLT2) inhibitors as a novel therapeutic approach to treating HFrEF patients, irrespective of the presence or absence of type 2 diabetes (T2D) [37,38,39,40]. However, the risk to patients remains high, and further studies are required to define new pharmacological approaches [34,35,36]. With regard to HFpEF, the diagnosis of which is more complex, up until now, clinical studies have not shown the existence of an effective therapy, but treatment is based on the use of diuretics and mineralocorticoid receptor antagonists, and exercise training is recommended [41,42,43,44]. At the moment, clinical studies aim at a better pathophysiological characterization in order to identify individual therapies [42,45]. Furthermore, self-care is very important for the management of HF patients. In particular, the Heart Failure Association (HFA) guidelines focus on maintaining self-care, consisting of regular drug intake and eventual adaptation to changes in the course of the disease. Maintaining self-care also includes a special focus on physical activity and diet. Self-care monitoring and management is also important as well as travel, leisure, smoking, sleep, immunization, and infection prevention [46]. Alongside pharmacological treatment, a non-pharmacological approach could be effective, especially in the early stages of the disease. In particular, it has been shown that many HF patients have a low intake of micronutrients, a condition that could trigger heart failure [5,47]. Therefore, in the early stages of disease onset, the intake of many plant extracts with antioxidant properties, such as bergamot polyphenolic fraction (BPF) extract, could counteract HF development [5,48]. In addition, both studies in animal models and in patients with HF indicate that the intake of n-3 polyunsaturated fatty acids (PUFAs) inhibits the onset of interstitial fibrosis and diastolic heart dysfunction [49]. In fact, many researchers are currently studying the use of natural compounds in the prophylaxis and treatment of various diseases, as there are studies that demonstrate their effectiveness in heart diseases, hepatic steatosis, metabolic syndrome, and neurodegenerative diseases [5,50,51,52,53,54,55].

For these reasons, in this review, we want to highlight the most recent studies related to the effect of PUFA supplementation in heart failure.

## 2. Heart Failure and PUFAs

PUFAs are long-chain polyunsaturated fatty acids with a carboxyl group (-COOH) at the polar hydrophilic end and a non-polar hydrophobic methyl group (-CH3) at the opposite end [56]. They are divided into two classes, n-3 and n-6 PUFAs, whose precursors are, respectively, α-linolenic acid (ALA, 18:3, Omega-3) and linoleic acid (LA, 18:2, Omega-6), which are defined as essential, as they must be ingested through dietary means [57,58]. ALA, present in beans, nuts, and flax seeds, is the metabolic precursor of eicosapentaenoic acid (EPA, 20:5 n-3) and docosahexaenoic acid (DHA, 22:6 n-3), while γ-linolenic acid and arachidonic acid (AA) derive from LA, which is mainly present in soy and corn oils [58,59]. Despite the synthesis pathway in the liver, the highest amount of EPA and DHA is achiever through diet, as they are present in foods of animal origin but, above all, in the oil and flesh derived from marine fish [56,57,58,59]. As far as their function is concerned, the n-6 PUFAs and their derivatives have pro-inflammatory action, unlike the n-3 PUFAs and their derivatives, which are powerful anti-inflammatories; therefore, the recommended daily intake of PUFAs is in favor of the latter [59,60,61,62]. Meta-analyses of prospective cohort studies indicate that fish consumption is inversely associated with HF risk. On the contrary, the risk of developing HF increases with a higher consumption of meat; therefore, fish can be considered a healthy animal-based dietary source of protein. Obviously, the positive association is also linked to the type of fish consumed and the cooking method. However, fish is considered to be one of the best cardioprotective foods. In interventional studies, it has been reported that the increased consumption of fish or the integration of n-3 PUFAs in diet has a protective action against cardiometabolic risks such as hypertension, inflammation, oxidative stress, and endothelial dysfunction in addition to having lipid-lowering properties [49,63]. It is interesting to underline that the hypotriglyceridemic, anti-inflammatory, anti-arrhythmic, and anti-thrombotic effects of n-3 PUFAs can be attributed to a modification of the DNA methylation profile in blood leukocytes [64]. Furthermore, both in vivo and clinical studies report the preventive action of n-3 PUFAs on cardiotoxicity and HF, which arise as side effects of anti-cancer therapies [65]. Thanks to their properties, n-3 PUFAs reduce ventricular remodeling and myocardial fibrosis, improving systolic and diastolic ventricular function altered in HF [66]. Furthermore, muscle wasting (understood as loss of muscle mass and strength) and cachexia (understood as weight loss) affect many HF patients, and supplementation with n-3 PUFAs appears to have a beneficial effect on these comorbidities. In fact, both the European Society of Cardiology (ESC) and the American College of Cardiology/American Heart Association (ACC/AHA) guidelines state that the integration of n-3 PUFAs into the diets of HF patients can be considered an additional therapy and is effective in reducing the death and hospitalization risk for these subjects [67]. Additionally, the AHA recommends eating 1–2 seafood meals per week, especially non-fried seafood [68,69]. In the Australian heart failure guidelines, of all the nutraceuticals considered, only the n-3 PUFAs have a recommendation, even though it is minimal [70]. Some clinical trials did not demonstrate the efficacy of treating HF with n-3 PUFAs, particularly with EPA; this could be caused by the failure to achieve a therapeutically effective concentration in all subjects [71]. Conversely, n-6 PUFAs can be cardiotoxic by acting directly on the heart muscle. In fact, the n-6 PUFAs, while reducing cholesterol levels and showing anti-atherosclerotic properties, can reduce survival after cardiac remodeling; this has been demonstrated in both in vitro and in vivo studies. These studies have shown that LA, but not derivatives such as γ-linolenic acid or arachidonic acid, stimulate the transforming growth factor-β (TGFβ) isoforms by increasing the collagen I/III ratio and lysyl oxidase (LOX) activity, causing an impaired transmitral flow that generates a cardiac “stiffening” with early diastolic dysfunction and normal systolic function [72].

### 2.1. Cardioprotective Mechanisms of n-3 PUFAs

Cardiac fibrosis, mainly determined by oxidative stress, is one of the most important causes of MI and HF. The nuclear factor erythropoietin 2 related factor 2 (NRF2) is involved in the process of controlling oxidative stress; due to its antioxidant activity, the NRF2 could protect the heart from the onset of cardiac fibrosis. Therefore, it is interesting that EPA, DHA, and some specialized pro-resolving lipid mediators (SPMs), such as resolvin D1 (RvD1), are capable of activating NRF2, demonstrating efficacy against cardiac fibrosis and therefore demonstrating effectiveness against HF and MI onset [73]. Furthermore, some mechanistic studies propose a different cardioprotective mechanism of n-3 PUFAs that involves their interaction with the free fatty acid receptor 4 (Ffar4), a G-protein coupled receptor (GPR) for long-chain fatty acids. The idea of the existence of this action mechanism derives from results obtained in a mouse model of pressure overload-induced HF, in which EPA counteracted the fibrosis onset without, however, incorporating itself into the cardiac myocyte and fibroblast membranes [71]. As far as Ffar4 is concerned, it is present both in isolated mouse cardiac myocytes and fibroblasts; it has also been isolated from hearts from rats fed a high-fat diet, but its expression in the human heart is not certain. Data obtained in primary cardiac fibroblasts indicate that Ffar4 inhibits fibrosis; moreover, it is a receptor for n-3 PUFAs. Therefore, it is possible to deduce that EPA inhibits fibrosis by interacting with Ffar4 without necessarily having to incorporate itself into membranes but by activating the anti-fibrotic endothelial nitric oxide synthase (eNOS)/cyclic guanosine monophosphate (cGMP)/cGMP/protein kinase G (PKG) signaling pathway [71]. Nutraceutical supplementation appears to have also yielded good results in patients with ventricular hypertrophy (VH)/HF syndrome. In particular, a low-fat diet that is particularly rich in EPA protects VH/HF patients from cardiac fibrosis by activating the fibroblast GPR120 receptors. Such a diet increases both the EPA and DHA content at the membrane level and the EPA/AA ratio. In this way, VH/HF patients are more protected from both sudden death arrhythmias and from coronary plaque formation [74] (Figure 2).

EPA activates the NRF2 (nuclear factor erythropoietin 2 related factor 2), Ffar4 (free fatty acid receptor 4) receptor, or the GPR120 fibroblast receptors. This activation inhibits cardiac fibrosis and protects the heart from the HF onset.

Furthermore, in a study conducted on a small number of patients (31 subjects) with ischemic HF who were treated with 2 g of n-3 PUFAs per day for a period of 8 weeks, the evaluated parameters indicate that the n-3 PUFAs increased the LVEF. In addition, both the global longitudinal strain and the early ventricular filling to the early mitral annulus velocities (E/e’) ratio are reported to have decreased. The levels of soluble interleukin-1 receptor-like 1 (sST2), an established marker of inflammation and myocardial fibrosis, and high-sensitive C Reactive protein (hsCRP) also decreased, while flow-mediated dilation increased. All of this indicates an improvement in inflammation, fibrosis, endothelial function, and ventricular systolic and diastolic performance as a consequence of the n-3 PUFAs’ actions. Obviously, given the limited number of patients, these results must be further verified and should also be verified in relation to the n-3 PUFA dose that was administered [75]. To try to understand the EPA effect on the free fatty acid increase that generates a cardiac lipotoxicity condition with impaired cardiac function following chronic adrenergic stimulation, a study was conducted on differentiated H9c2 myocytes. These cells were treated with 400 μM of palmitate (saturated fatty acid) for 24 h. Following the treatment, an increase in ROS production, a reduction in the ATP synthesis, and an increase in the apoptotic process were recorded. Palmitate stimulates dynamin-related protein-1 (Drp1) expression, which increases mitochondrial fragmentation. On the contrary, in the presence of EPA (50 μM), palmitate-induced apoptosis and mitochondrial dysfunction as well as Drp1 expression and mitochondrial fragmentation are reduced following AMPK (AMP-activated protein kinase) activation. Through these mechanisms, the EPA is able to protect myocytes from lipotoxicity and therefore is able to prevent HF aggravation [76]. In addition, in a small, double-blind, placebo-controlled, randomized study, the effect of n-3 PUFA supplementation on the high-density lipoprotein (HDL) antioxidant capacity was investigated. This study arose from the knowledge of the oxidative stress involvement in HF pathogenesis [77]. In fact, the association between the HDL antioxidant properties, HF, and clinical outcome was demonstrated. In addition, in some studies, an increase in HDL levels has been reported in HF patients following integration with n-3 PUFAs, with a consequent reduction in the risk of cardiovascular events (CVD) [77]. In this study, 40 subjects with advanced HF of non-ischemic origin, New York Heart Association (NYHA) class III or IV, who had been on therapy for three months or more were treated with 1 or 4 g of n-3 PUFAs per day for 12 weeks. The obtained results, however, indicate that in this cohort of patients, supplementation with n-3 PUFAs reduced the HDL antioxidant capacity. It should be noted that in this study, only patients with non-ischemic HF of NYHA classes III or IV were treated, so it should be investigated as to whether the same results would also be obtained with other HF populations. Therefore, the contrasting effect of n-3 PUFA supplementation on the antioxidant capacity of HDLs could be related to the severity of the inflammatory condition [77]. In addition, recently, a study was conducted in which cardiac hypertrophy was generated by ligating the aortic arch, resulting in pressure overload in C58BL/6J mice. In this condition of cardiac hypertrophy and dysfunction, the fatty acid composition in the plasma and the heart was evaluated, recording a higher concentration of free AA and its pro-inflammatory derivatives. Treatment with Omega-3 EE, a clinical drug consisting of both EPA and DHA, resulted in an improvement. In particular, in the 5 days preceding the operation and until the end of treatment, the mice were administered Omega 3-EE at 1.5 mg/g of body weight daily. This resulted in the inhibition of pro-inflammatory cytokine expression, while a reduction in the inflammatory cell infiltration into the heart and a reduction in cardiac dysfunction and fibrosis were also recorded. These data allowed the conclusion that n-3 PUFAs prevent HF onset, as they resolve the inflammatory process, not inhibiting the initial phase but resolving the subsequent phases [78]. It is also known that hypertension is one of the most important risk factors for the HF onset, and diet has been shown to affect its development [79]. In fact, a diet rich in n-3 PUFAs was found to be effective in reducing diastolic and systolic blood pressure values; therefore, a diet enriched with fish is recommended. Excessive fish intake, however, could lead to a buildup of heavy metals, such as mercury [79]. Therefore, the focus has been on the intake of new generation chicken eggs that can be obtained by enriching the diet of laying hens with PUFAs. In this way, eggs are obtained with yolks containing lecithin (SL) esterified with n-3 and n-6 PUFAs. These eggs were used to treat hypertensive rats (SHR). The data obtained from that study indicate that the SL derived from these eggs reduces the values of the systolic and diastolic pressure in addition to the inflammatory state and arterial vasoconstriction. In addition, a lower heart rate and a reduction in oxidative stress have been reported in both normotensive and hypertensive rat strains. The knowledge regarding the positive effects of n-3 PUFAs on both arterial blood pressure and on coronary blood vessels as well as on vasodilator prostaglandins synthesis and on the reduction in pro-inflammatory cytokines synthesis allows us to conclude that the results obtained in the aforementioned study are attributable to a combination of the SL with the n-3 PUFAs [79].

### 2.2. Effect of EPA/DHA Supplementation in HF

Important results were derived from studies in which the effect of EPA/DHA supplementation was evaluated in HF or other CVD rather than from studies deriving from dietary advice on the intake of common foods with different contents of EPA and DHA. This different approach allows researchers to monitor the dosage effect, determining the dose–effect relationship [80]. In this regard, Bernasconi A.A. et al., 2021, considered 40 studies with a total of 135,267 participants [80]. This allowed the authors to establish both the efficacy of n-3 PUFA supplementation in preventing CVD events, such as HF and MI, and their greater protective effect in the presence of a higher dosage. In particular, in this study, the beneficial effects were associated with EPA and DHA supplementation corresponding to dosages from 1 to 2 g/day, values that cannot be obtained only with a diet rich in fish [80]. The efficacy of n-3 PUFA supplementation, especially in the early stages of HF, has been proven by the results of clinical studies such as the GISSI-HF trial, which lasted 3.9 years, in which the 6975 patients who were enrolled had chronic HF of NYHA class II-IV. Patients treated with 1 g/day of n-3 PUFAs showed a greater increase in baseline LVEF over 3 years compared to the placebo group. In addition, a reduction in cardiac death was reported in the EPA and DHA groups compared to the control; this was more relevant in the groups treated with an EPA + DHA dosage > 1 g/day. Therefore, it is evident that treatment efficacy is related to the dosage and duration [81]. These positive results related to the n-3 PUFA intake were confirmed in other studies, leading the European Society of Cardiology to conclude that supplementation with EPA and DHA reduces both hospitalization and death from cardiovascular complications, i.e., an improvement in HF prognosis was found [81]. Furthermore, a randomized clinical trial was conducted on 205 patients suffering from chronic compensated heart failure, determined by ischemic (IHF) or dilated cardiomyopathy (DCM)—NYHA classification I–III. For six months, the diet of half of these patients was supplemented with 1 g of n-3 PUFAs. The clinical picture resulting from the echocardiography that was performed as well as the plasma brain natriuretic peptide (BNP) and creatinine levels indicates a positive effect of PUFAs on cardiac hemodynamics. In particular, left diastolic function and circulating BNP levels improved, resulting in a positive effect on end-diastolic pressure. These data are relevant for long-term prognosis due to the effect obtained on ventricular filling pressures, right ventricular function, and renal function [82].

In addition to this, another meta-analysis of randomized controlled trials confirmed lower levels of BNP and serum norepinephrine (sNE) following n-3 PUFA supplementation in HF patients [83]. A study in patients with decompensated heart failure reported a significant depletion of EPA and DHA even though this was not correlated with an increased three-year mortality risk [84]. Despite the contradictory data on the benefits and optimal doses of n-3 PUFAs, which are very often linked to the design and execution of the study, it seems sufficiently clear that n-3 PUFAs can be used in preventive cardiology. In fact, they show modest but statistically significant positive effects with a progressive trend over time [85].

### 2.3. PUFAs in HFrEF and HFpEF

The Multi-Ethnic Study of Atherosclerosis (MESA) study was recently conducted; it was the first clinical trial conducted to define the role of n-3 PUFAs in the primary prevention of HF incidence. There were approximately 6562 participants in the MESA cohort (52% women), including African Americans, Hispanics, Asians, and whites from the United States, who were aged 45 to 84. The median follow-up period was approximately 13 years. In particular, plasma EPA abundance was measured, which was expressed as %EPA [86]. Previous studies in mice, in a model of pressure overload-induced HF reproducing HFpEF remodeling, showed that EPA prevents contractile dysfunction and fibrosis in a concentration-dependent manner. In light of this, correspondence was also sought in humans [86]. Therefore, in the MESA study, Block R.C. et al., 2019, established that a higher plasma %EPA is associated with a lower risk of HF, including HFrEF and HFpEF, and this is also valid for the other n-3 PUFAs, especially for the %EPA + %DHA combination [86]. Furthermore, there are many studies that have been conducted on patients with HFrEF that demonstrate the beneficial effects of treatment with n-3 PUFAs on heart rate (HR) and on the reduction in the serum level of BNP, resulting in effective therapy for ventricle wall tension [87]. This confirms that n-3 PUFA supplementation reduces heart failure-related hospitalizations and death in patients with HFrEF [88]. Regarding HFpEF, it is known to be associated with obesity; therefore, clinical studies have been conducted in obese patients with HFpEF and in mice with diet-induced cardiac dysfunction in order to evaluate the effect of unsaturated fatty acids (UFA), including monounsaturated fatty acids (MUFAs) and PUFAs. Clinical results have shown that a diet rich in UFA (MUFAs and PUFAs) determines an improvement in diastolic function and cardiorespiratory fitness (CRF) and corresponds with an increase in fat-free mass. Even in the mouse model, the diet enriched in UFAs and with a reduced sugar intake limited weight gain and prevented diastolic dysfunction [89].

Additionally, Matsuo N. et al., 2021, conducted a retrospective single-center cohort study [90]. The study included 140 hospitalized patients with acute decompensated HFpEF. In these subjects, the plasma levels of EPA, DHA, AA, and dihomo-gamma-linolenic acid (DGLA) were evaluated. The obtained data showed that low DHA levels are associated with an increase in all-cause death. This indicates that DHA plasma levels may not only be useful in diagnosis but may also be the target of the therapies administered to such patients [90].

### 2.4. EPA/AA Ratio and HF

The clinical relevance of the EPA/AA ratio related to the mortality of hospitalized HF patients was studied in 577 subjects hospitalized for HF who were divided into 2 groups: one comprising subjects with EPA/AA ratios ˂ 0.32 mg/dL and another with EPA/AA ratios ≥ 0.32 mg/dL. The two groups did not differ in some parameters such as blood pressure, BNP levels, and LVEF. On the contrary, they differed in cardiac mortality, which is inversely proportional to the EPA/AA ratio, indicating that this ratio can be considered an independent predictor of cardiac mortality in HF subjects. This finding is more evident in HF patients who take statins. Therefore, the EPA/AA balance could be considered to try to reduce the mortality risk in HF [91]. Furthermore, in a trial conducted on 130 adult patients with congenital heart disease (CHD), the EPA/AA ratio in association with adverse cardiac events was evaluated over a follow-up period of approximately 15 months. It was established that a low EPA/AA ratio was associated with a greater risk of hospitalization for HF; therefore, the EPA/AA ratio can be considered a good predictor of cardiac events in adult patients with CHD [92]. It is reported that in chronic HF, a higher EPA/AA ratio corresponds to a condition with reduced inflammation and improved endothelial and cardiac function [93].

### 2.5. PUFAs and ADHF

The roles of EPA, DHA, AA, and DGLA have also been studied in acute decompensated HF (ADHF) [94]. In particular, the PUFA dosage was evaluated by Nagai T. et al., 2016, in 685 patients hospitalized for ADHF, with a follow-up period of approximately 560 days, considering all-cause death and aggravation of HF as adverse events [94]. Dosages demonstrated that low levels of n-6 PUFAs were present in patients experiencing more severe clinical outcomes. Conversely, there appears to be no correlation between EPA and DHA levels and adverse events. An explanation for the low n-6 PUFA levels in patients hospitalized for ADHF could be linked to the pro-inflammatory role of these fatty acids which therefore could be consumed more often following a more consistent inflammatory response. Decompensated HF also presents a lower synthesis of the potent vasoactive agents, prostaglandin E1 (PGE1), prostacyclin (PG12), and NO, as a consequence of the low AA and DGLA levels. Furthermore, in these patients, the consequences of advanced HF can lead to a condition of malnutrition, which also affects the n-6 PUFA supply. Therefore, the dosing of n-6 PUFAs in ADHF patients could provide information on the risks these patients face after hospitalization [94]. In another study on ADHF by Ouchi S. et al., 2017, over the course of about 2 years, 267 hospitalized patients suffering from this disease were monitored [95]. In particular, the evaluation of the geriatric nutritional risk index (GNRI), which is representative of the nutritional status of patients, associated with long-term mortality and PUFA levels allowed the authors to draw some conclusions. Indeed, low levels of DHA, DGLA, and AA were independently associated with long-term mortality across various nutritional statuses in this study. The same association was not found for EPA. This difference between EPA and DHA is probably due to the different levels of the two fatty acids in the cardiocyte membranes; in fact, DHA is abundant, as opposed to EPA, thus influencing cell membrane function. Additionally, this study indicated that a the time of admission patients with ADHF present a condition of malnutrition, especially in relation to essential fatty acid intake [95]. Furthermore, it has been shown that in ADHF patients, not only low DGLA levels but also a low DGLA/AA ratio are associated with long-term mortality [96].

### 2.6. Effect of PUFAs in HF Development after MI

Advances in preventive cardiology related to HF were presented to the AHA in 2020. Among others, the Omega-3 Fatty Acids in Elderly Patients with Acute Myocardial Infarction (OMEMI) trial by Kalstad A.A. et al., 2021, was presented [97]. In this study,1027 elderly patients who had suffered an MI within the 2–8 weeks prior to the study were treated with 1.8 g of a EPA + DHA combination or with a placebo based on corn oil, with a follow-up period of 2 years [97]. Over this time frame, the death rate from MI-related events was similar in both PUFA and placebo-treated subjects. Therefore, the study failed to demonstrate a clear positive effect of n-3 PUFAs in preventing adverse cardiovascular events that could occur in elderly MI patients, probably because the study was conducted for a relatively short period of time and on a relatively small sample. What was shown was an increase in atrial fibrillation in the presence of n-3 PUFAs, a fact that should be taken into account when these therapies are administered to these patients [97,98]. Furthermore, in a sub-study of the OMEMI trial, it was found that the levels of the cardiac remodeling marker, galectin-3, are lower in the presence of higher EPA and DHA levels, indicating a positive effect of n-3 PUFAs on this condition [99]. It is also known that diabetic individuals are more likely to develop HF following acute MI, as they have more predisposing factors for post-infarct adverse left ventricular remodeling. These factors include increased vascular stiffness, endothelial dysfunction, the upregulation of inflammatory mediators, and insulin resistance [100]. In this context, Fujikura K. et al., 2020, conducted a randomized clinical trial called OMEGA-REMODEL to better clarify the role of insulin resistance and the possible effect of n-3 PUFAs [100]. In this clinical trial, 325 acute MI patients were treated with 4 g/day of n-3 PUFAs for a period of 6 months. The evaluations that were conducted allowed the authors to establish that n-3 PUFAs reduce both cardiac remodeling and inflammation, even if insulin resistance influences this response. More precisely, in patients with lower insulin resistance, a better response to treatment with high doses of n-3 PUFAs was recorded, which was characterized by a reduction in the left ventricular end-systolic volume index (LVESVI) and therefore a greater reduction in adverse remodeling. It is very likely that this different effect is due to the presence of chronic inflammation in addition to acute inflammation in patients with high insulin resistance [100]. Moreover, a prospective cohort study, Pharmacogenomic Resource to Improve Medication Effectiveness Genotype Guided Antiplatelet Therapy (PRiME-GGAT) by Halade G.V. et al., 2020, was conducted to identify the differences in PUFAs and SPMs between Black and white patients following acute MI [101]. It is known that MI is a condition that determines HF onset (advanced HF post-MI), especially in Black subjects compared to the white population [101]. In this study, plasma was collected 24–48 h after MI from 22 Black (15 male and 7 female) and 31 white (23 male and 8 female) patients [101]. The assays conducted by liquid chromatography–mass spectrometry showed that the levels of AA and DHA were higher in white subjects of both sexes, while the level of EPA was higher only in white males [101]. The SPM family includes resolvin E1 (RvE1), which has a lower concentration in the Black population, as opposed to protectin D1 (PD1), which was lower only in white males. This study allowed the authors to evaluate the lipid species present during the inflammatory phase following MI. These results, together with the data relating to the higher incidence of heart disease in Black individuals, indicate that there is an important component to be traced back to lifestyle and nutrition, but above all, differences in pathophysiology related to race and sex were defined. The data found could help in defining a personalized therapy for HF management [101]. In order to understand the effect of EPA on post MI cardiac remodeling, a study was conducted on C57BL/6J mice treated with a high dose of EPA (1 g/kg), administered via gavage once a day for the 4 weeks before and in the 4 weeks following the experimentally induced MI by permanent coronary artery ligation. Following this long-term treatment, a reduction in mortality after MI was reported, especially in mortality due to HF, the main cause of death after MI. It has been shown that long-term treatment with EPA reduces HF onset after MI by acting on chronic cardiac remodeling, as EPA is capable of reducing the polarization towards pro-inflammatory M1 macrophages. Indeed, EPA shifts the balance from the activation of pro-inflammatory M1 macrophages to that of anti-inflammatory M2 macrophages [102].

### 2.7. n-3 PUFAs and Depression in HF Patients

Depression is known to be another one of the comorbidities affecting HF patients, resulting in a worsening of their health conditions [103]. Therefore, a preliminary study was conducted on a small number of patients suffering from chronic heart failure (CHF) and major depressive disorder. In the trial called Omega-3 Supplementation for Co-Morbid Depression and Heart Failure Treatment (OCEAN), Jiang W. et al., 2018, evaluated the effect of EPA and DHA supplementation on circulating levels of these PUFAs and on depressive disorders that severely afflict patients with CHF [104]. Patients were treated with both a combination of EPA + DHA in a 2:1 ratio and with EPA alone as well as a placebo, for a period of 12 weeks. The results indicate an increase in circulating PUFAs levels as well as an improvement in cognitive depressive symptoms, probably related to more favorable health conditions. These results need to be further investigated based on larger patient groups and for longer treatment times [104].

## 3. Conclusions

Around the world, there are numerous cases of hospitalization linked to HF; for this reason, there are many studies aimed at finding an effective therapy, especially a non-pharmacological preventive therapy. Clinical and in vitro studies indicate that the positive outcomes of dietary supplementation with EPA and DHA on the HF onset and management are related to the duration of treatment and dosage. It is clear that a diet with a more controlled PUFA intake can contribute to the improvement of inflammatory status, endothelial function, ventricular systolic and diastolic performance, inhibition of cardiac fibrosis, and remodeling in order to reduce hospitalization and death due to heart failure, including following MI (Figure 3). Studies on the use of PUFAs in HF need further confirmation and investigation, but current knowledge is already sufficiently favorable.

## Figures and Tables

**Figure 1 nutrients-13-02965-f001:**
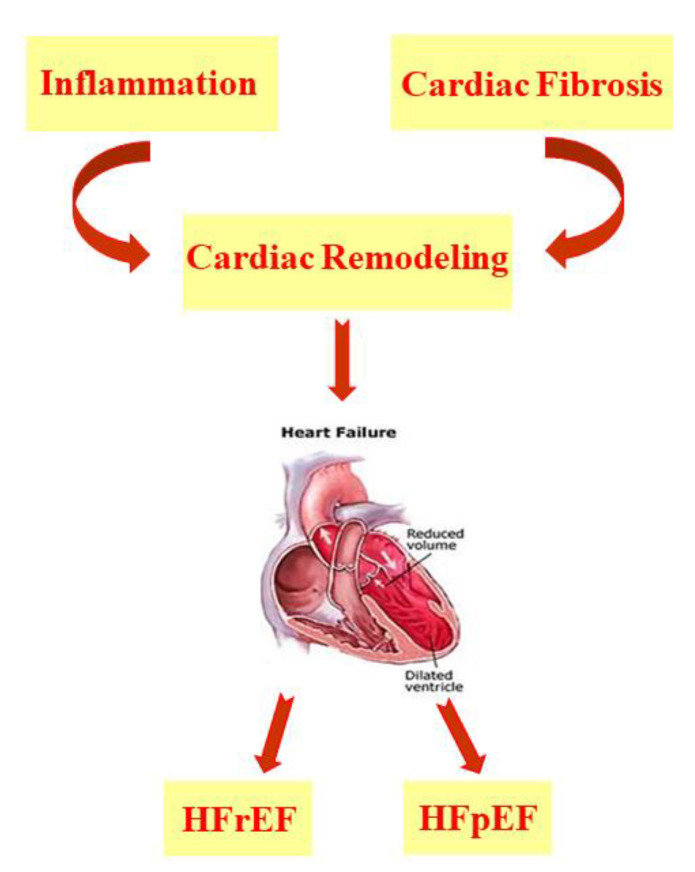
Characteristics of heart failure. HFrEF: patients with HF with a reduced ejection fraction; HFpEF: patients with HF with a preserved ejection fraction.

**Figure 2 nutrients-13-02965-f002:**
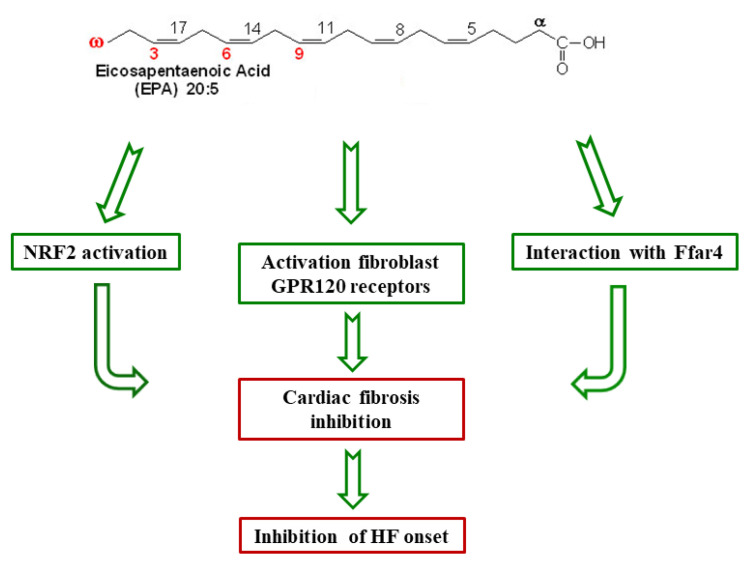
Cardioprotective mechanisms of eicosapentaenoic acid(EPA). Ffar4: the free fatty acid receptor 4; NRF2: the nuclear factor erythropoietin 2 related factor 2; GPR120: the G-protein coupled receptor 120; HF: heart failure.

**Figure 3 nutrients-13-02965-f003:**
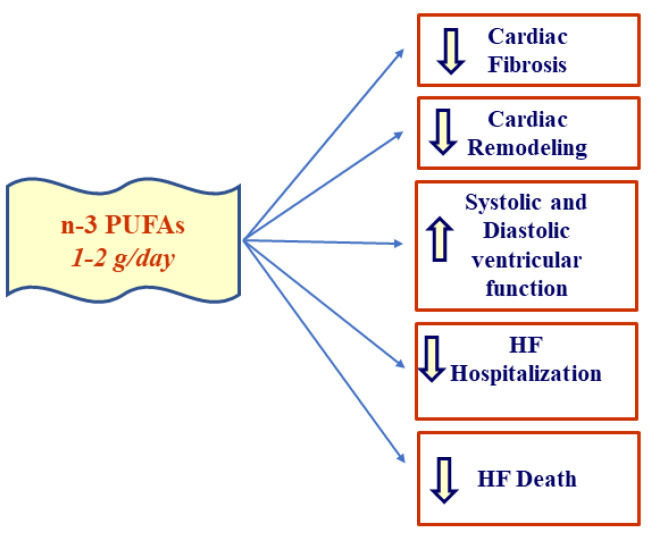
Effects of n-3 PUFAs supplementation in heart failure (HF).

## Data Availability

Data available in a publicly accessible repository.

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
