# Peer review of "PUFA Supplementation and Heart Failure: Effects on Fibrosis and Cardiac Remodeling"

_nutrients, 2021, doi:10.3390/nu13092965_

Round 1

Reviewer 1 Report

I thank you for giving me the opportunity to review this manuscript. I read it carefully and with interest about the review of the effects of supplemental PUFA intake in heart failure. The preventive effects of fibrosis and cardiac modeling after heart failure are discussed with the latest findings, and I believe the article describes the comprehensive knowledge and unresolved matters in this area. The writing and logic of the manuscript is appropriate, and the figures are also appropriate to aid understanding.

Author Response

We thank the reviewer for his positive comments.

Reviewer 2 Report

A good well written comprehensive review, however  I would suggest that the authors include their literature search strategy in a Methods section, in order to give confidence to the reader that all appropriate material is covered.

Few minor points should be addressed :

1.Page 4 line 135 should be present ; page 4 line 143 –… fish a good source of protein?- this should be clarified

  1. The authors mention several factors involved in the pathogenesis of heart failure and suddenly on page 5 line 212 they refer to sST2 without any explanation of its full name and a role as a biomarker. In the same sentence they write „highly sensitive CRP” which is a wrongly used termhsCRP means that CRP was measured with a high sensitivity method and presented in mg/L.
  2. On Page 7 line 282 the dose of PUFA is given in mg/day and later, line 286 in g/day ?; it would be much better to use the same units.

4.Finally on page 7  line 305 the abbreviation SNE is used for serum norepinephrine; it is generally accepted to  use sNE for serum derived metabolites.

5.I would suggest to use the same font in all the figures  (Times New Roman  or Arial)

Author Response

Response to Reviewer 2 Comments

Point 1: A good well written comprehensive review, however I would suggest that the authors include their literature search strategy in a Methods section, in order to give confidence to the reader that all appropriate material is covered.

Response 1: We thank the reviewer for the suggestion. In the abstract section the sentence " For that purpose, a PubMed literature survey was conducted with a focus on some in vitro and in vivo studies and clinical trials from 2015 to 2021." has been added.

Few minor points should be addressed:

Point 1: Page 4 line 135 should be presentpage 4 line 143 –… fish a good source of protein?- this should be clarified.

Response 1: The word “presents” has been changed to “present”; In order to clarify “–… fish a good source of protein” has been added “On the contrary, the risk of developing HF increases with a higher consumption of meat; therefore, fish can be considered a healthy animal-based dietary source of protein.”

Point 2: The authors mention several factors involved in the pathogenesis of heart failure and suddenly on page 5 line 212 they refer to sST2 without any explanation of its full name and a role as a biomarker. In the same sentence they write „highly sensitive CRP” which is a wrongly used term – hsCRP means that CRP was measured with a high sensitivity method and presented in mg/L.

Response 2: The sentence has been changed as suggested.

Point 3: On Page 7 line 282 the dose of PUFA is given in mg/day and later, line 286 in g/day ?; it would be much better to use the same units.

Response 3: The unit has been changed as suggested.

Point 4: Finally on page 7 line 305 the abbreviation SNE is used for serum norepinephrine; it is generally accepted to use sNE for serum derived metabolites.

Response 4: The abbreviation has been changed as suggested.

Point 5: I would suggest to use the same font in all the figures (Times New Roman or Arial)

Response 5: The font has been changed and the figures replaced.